# Research on Sustainable Mining and Water Prevention in Large Open-Pit Water Deposits

Yalei Zhe [1,*] , Kepeng Hou [1], Wei Liang [2] and Huafen Sun [1,*]

1 Faculty of Land Resources Engineering, Kunming University of Science and Technology, Kunming 650093, China; 11301046@kust.edu.cn
2 Institute of Mining Engineering, Guizhou Institute of Technology, Guiyang 550003, China; 20160713@git.edu.cn
* Correspondence: 20201101026@stu.kust.edu.cn (Y.Z.); 20140091@kust.edu.cn (H.S.)

**Abstract:** Due to the impacts on the ecological environment, production safety and the economic benefits of large open-pit water deposits, there is no longer a single drainage method within the pit that can be used to meet the need for further deep mining. Therefore, curtain grouting technology is proposed for use in the prevention and control of mine water. The flow control equation of slurry in vertical fissures under the influence of self-weight stress was derived, and a three-dimensional visualized laminated jointed rock grouting model was developed independently to study the slurry transport mechanism, diffusion pattern, and sealing mechanism, which verified the correctness of the control equation. Field trials of industrial curtain grouting were also carried out at the mines, and the effectiveness of curtain water blockage was tested using visual analysis, inspection hole detection, and geophysical exploration methods. The results showed that after grouting the curtain, water-conducting cracks in the formation can be filled well and sealed, and the curtain has a good water-blocking effect. A complete set of diversified value-taking grouting systems is obtained, which can ensure the development of a green, safe and sustainable mine in the future.

**Keywords:** large open-pit water deposits; flood control; sustainable mining; curtain grouting; grouting parameters





## 1. Introduction

Hydraulic disasters are a well-recognized problem that often occurs in subsurface mining operations. Due to the peculiar nature of the enclosed and narrow subterranean space, hydrologic disasters are sudden and serious, with consequences that are difficult to address after the event [1]. For this reason, researchers have carried out extensive studies on techniques for water control in underground mines with the aim of obtaining the ability to achieve zero mine water discharge through an optimal combination of mine water control, treatment, utilization and re-injection [2]. For instance, Hugo conducted research on groundwater control techniques in the strong karst environment of the Vazante underground mine in Brazil [3]; Li et al. [4] analyzed the water inrush mechanism caused by the failure of the aquifer and the overlying confined karst aquifer by studying the activation of faults in the process of mining. Li et al. [5] used numerical simulation methods in combination with experiments on the discharge of water from the subsurface in order to predict the amount of drainage and depressurization, and developed a drainage and depressurization scheme. Yin [6] proposed a prediction method based on anomalous analysis of water level changes in groups of boreholes using a coupled long-and short-term memory network (LSTM) and isolated forests model and set out a set of classification rules for the hazard of water inrush.

However, in recent years, the sustainable mining problems of many large open-pit water deposits have also been profoundly affected by water catastrophes, especially in mines with sunken open-pit mines [7,8]. The term 'large open-pit water deposit' refers to

a mine that has a high water content in the aquifer at the site of the mine, and the inflow of water from the mine shaft reaches tens of thousands of cubic meters per day, in which proactive measures need to be taken to control and prevent the inflow of water into the mining pit [9–11].

As mining deepens, the inflow of water into the pit of a large sunken open-pit water deposit will gradually increase, and this will have a large effect on the mining operation. Long-term slope immersion in water can also pose a threat to the stability of the slope [12,13]. Conventional methods of controlling water storage include drainage and precipitation, but large amounts of extraction and drainage may disrupt the groundwater balance, leading to the drying up of surrounding wells and hindering the irrigation capacities of villages. This is a serious threat to the ecological security of the mining area, which has led to geological and mining conflicts [14–18]. This is not something that mining production wants to see, and as the depth of mining increases, the inflow of water will rise significantly. The mine shaft extraction and drainage engineering process consumes a large amount of personnel and equipment, which costs tens of millions of yuan and imposes a serious economic burden on the production of mines, raising the cost of surface mine production and thus reducing corporate profits [19]. Moreover, simple drainage can no longer satisfy the needs of mining deeper into the mines. As a result, a large number of researchers have borrowed from the experience of dam grouting and water blockage in hydraulic engineering [20] and proposed the use of curtain grouting technology for the prevention and control of water works in mines. Yuan et al. [21] summarized and analyzed the current state of curtain grouting technology in deep underground mines based on four aspects: curtain construction conditions, theoretical design and efficiency, borehole structure, and the development of grouting materials. Liu et al. [22] based on the numerical manifold method (NMM), proposed an NMM-HM grouting model is proposed to investigate slurry flowing and analyze grouting efficiency. Ren et al. [23] used adjacent ore body curtain grouting technology in Laixin Iron Mine to control groundwater, leading to significant economic and environmental benefits. A curtain parameter survey was successfully applied by Li et al. [24] in the complicated hydrogeological conditions at Zhongguan Iron Mine. Cai [25] proposed a novel evaluation model using an analytic hierarchy process and ideal point method to study the grouting effect in a weak and water-rich stratum.

It can be seen that a large number of scholars have studied the grouting of single horizontal fissures, and there are fewer studies on multiple interconnected fissures. Therefore, in this paper, we have studied the transport mechanism, diffusion morphology and sealing mechanism by flow control equation theoretical analysis and indoor similar model test for the characteristics of laminated jointed rock masses in mines, and conducted a systematic analysis of grouting parameters using a field large-scale grouting test with various testing means, which can provide a reference basis for the water control work in similar open-pit large water mines.

## 2. Overview of Mines

The Jianshan Phosphate Mine is a site belonging to a marine sedimentary deposit consisting of layered phosphate rock, mainly hosted within the second Lower Cambrian section of the Meishucun Formation. The ore bed slopes and runs south to north and northeast in a 30° direction. It is remarkable that the ore layers occur in fixed layers, and that their distribution is consistent with strata, basically stratified or quasi-stratified. The Jianshan Phosphate Mine adopts an open-pit mining method. Results from multiple hydrogeologic explorations indicate that hydrogeologic conditions less than 1910 m in the Jianshan Phosphate Mine surface mining area are complex, with an initial increase in groundwater level of 1910 m. The mine has now been worked up to 1875 m, with a large inflow rate of water of 80,000 $m^3$/d. Due to the environmental requirement limits of the drainage into and out of the pit, d multiple accumulation basins were formed at the bottom of the pit during the production process along the strike of the ore body. As shown in Figure 1, the Jianshan Water Basin is the largest of the accumulation basins. The basin is

more than 10 m deep, with an accumulated water volume of approximately 137 m³, which severely restricts sustainable mining of the lower part of the mine.

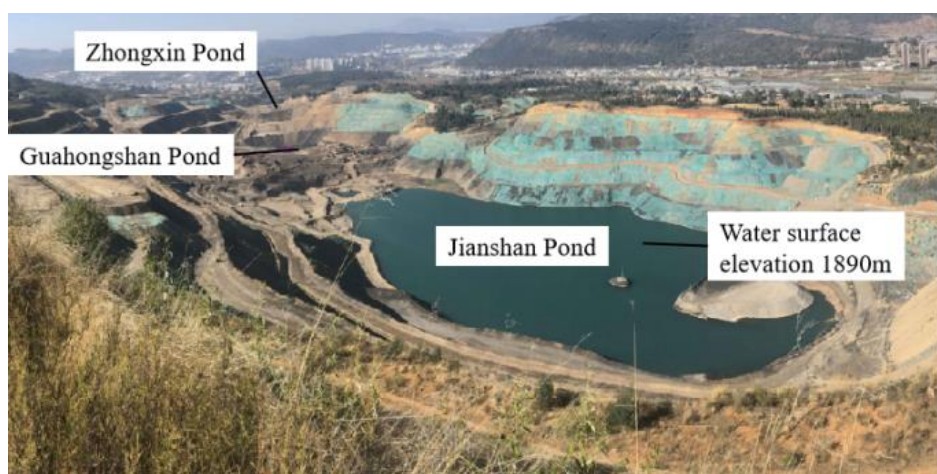

**Figure 1.** Water accumulation in the mine pit.

### 3. Grouting Simulation Test for Layered Jointed Rock Mass

Multiple factors affect the effect of grouting, one of which is the slurry diffusion law, i.e., how the slurry diffuses once it has passed through a fixed crack opening, and what migratory mechanism emerges as the slurry is continually injected. Research into the diffusion and sealing mechanism of single-crack grouting has been carried out based on fluid dynamics theory. A one-dimensional simulation test system for visual infiltration grouting was designed by Zhang [26] to obtain variation in the cement–water glass slurry (C-S) grouting pressure as a function of time under different media permeability conditions and different grouting flow rates. Gustafson et al. [27] derived an equation for the transport of slurry for a single plate fracture under constant-pressure grouting conditions on the basis of the Bingham body constitutive model. Hassler et al. [28] considered the slurry to be a Bingham fluid and derived the diffusion equation for the fracture slurry.

However, there is a lack of research on the diffusion law of multiple fractures, especially the interconnectedness between multiple layered and vertical fractures, and there is a lack of large-scale model testing or on-site testing to analyze and verify the diffusion law of slurries of cement injected into jointed rock masses in three dimensions. In order to obtain more accurate and convenient scattering patterns and to assess the suitability of the slurry properties in geological bodies under specific grouting parameters, experimental studies of large-scale physics simulations are an effective means of addressing the above issues. To this end, a three-dimensional visualization model of the jointed rock mass grouting is developed independently on the basis of the geologic characteristics of the jointed mass of bedded rock at the Jianshan Phosphate Mine. The scattering mechanism of spatially connected fracture growth has been widely explored, and suggestions for improvement are offered for the preliminary design of the traditional grouting process on the basis of this.

#### 3.1. Slurry Flow Control Equation

To organize the slurry diffusion control equation, Wei et al. [29] used the general expression for the slurry diffusion control equation in flat cracks as follows:

$$\frac{dp}{dr} = k_1 \frac{q\mu(t)}{rb^3} + k_1 \frac{\tau_0}{b} \tag{1}$$

where $P$ is the grouting pressure, $r$ is the diffusion distance of the slurry, $q$ is the grouting rate, $b$ is the crack width, $\mu(t)$ is a time-varying function of the slurry viscosity, and $\tau_0$ is the initial yield stress of the slurry. From Equation (1), it can be seen that the pressure at

a certain point in the crack is inversely proportional to the diffusion radius; that is, the further the slurry diffuses, the greater the pressure difference, as shown in Figure 2.

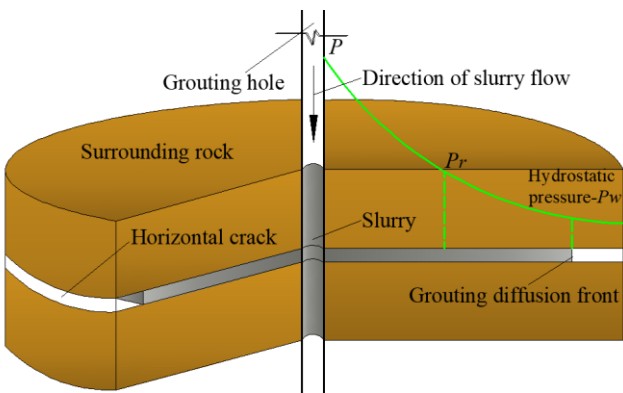

**Figure 2.** Schematic diagram of diffusion pressure distribution of slurry in horizontal fractures.

When the vertical fissure is connected with the horizontal fissure, the grout enters the vertical fissure from the horizontal fissure and climbs upward. The grout must overcome not only the shear stress, but also the gravity of the grout itself. The research confirms that the water–cement ratio is 0.8~1. The cement slurry of 0 conforms to the constitutive model of Bingham fluid, as follows:

$$\tau = \tau_0 + \mu\gamma \tag{2}$$

where $\tau$ is the initial yield stress of the slurry and $\gamma$ is the shear rate. The pressure exerted on the bottom of the vertical crack by the slurry is:

$$p_y = \tau_0 + \mu\gamma + \rho g\overline{h}bdr \tag{3}$$

When in the vertical fissure, the slurry diffusion velocity is 0, which means that when it rises to the maximum height, the direction of the slurry shear stress changes to the same direction as the grouting pressure, which jointly overcomes the gravity of the slurry, as shown in Figure 3, with the following expression:

$$p_y + \tau_0 = \rho g\overline{h}bdr \tag{4}$$

where $\rho$ is the slurry density, $g$ is the gravitational acceleration, and $\overline{h}$ is the height of the slurry in the vertical fissure when the horizontal diffusion radius increment of the slurry is $dr$.

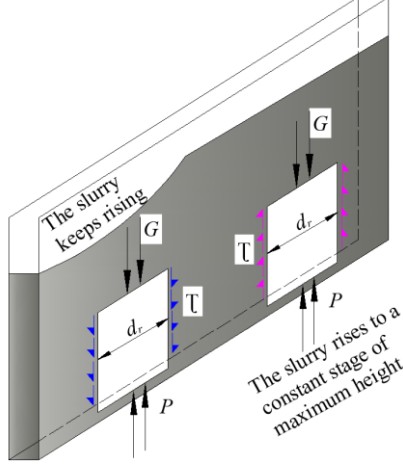

**Figure 3.** Schematic diagram of the forces on the slurry in the vertical fissure.

Combining Equations (1) and (4), it can be seen that the rising height of the slurry in the vertical fissure shows a decreasing trend with the increase in the horizontal diffusion radius.

### 3.2. Test Setup

The grouting borehole is a vertical downward hole and there is a certain distance between the two holes. Compared with the vertical fractures of each anisotropy, the grouting borehole can more easily expose the laminar fractures directly, and the slurry in the borehole will be pressed into the laminar fractures directly, while for the vertical fractures not directly exposed by the borehole, the slurry will be filled by the laminar fractures as a transmission hub for diffusion. According to this idea, the model is processed and produced. The test device consists mainly of three parts: ① is the frame of the test bearing platform; ② is a fracture system connected horizontally and vertically; and ③ simulates a drilling and grouting pipeline with a given length of the grouting section. For visualization of slurry grouting and diffusion, both the drilling slots and the grouting pipelines are composed of 1.2 cm-thick transparent acrylic plates. The crack space is achieved by adding a 3 mm spacer between two plates, simulating a crack channel with both vertical and horizontal dimensions of 3 mm. Drawings of the device's design are as in Figures 4 and 5:

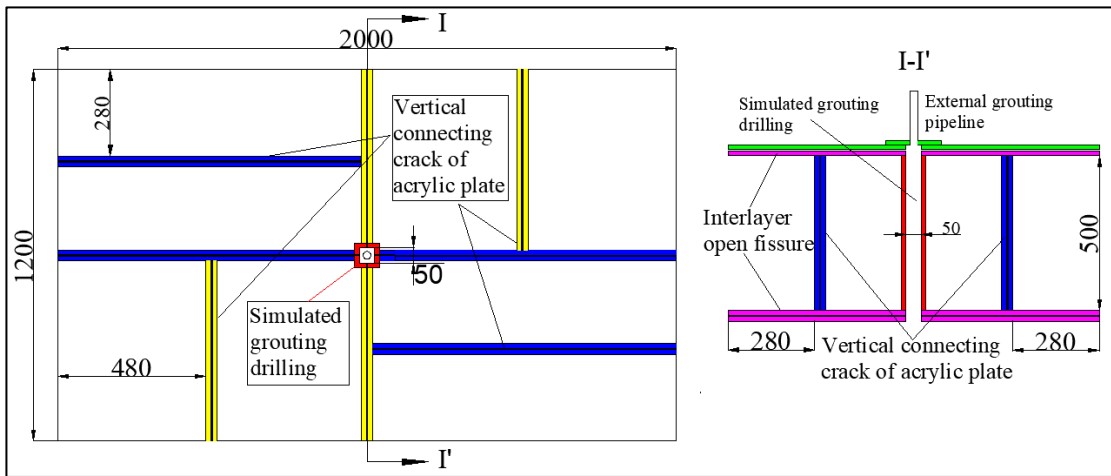

**Figure 4.** Design diagram of three-dimensional visualization layered jointed rock mass grouting model (unit: mm).

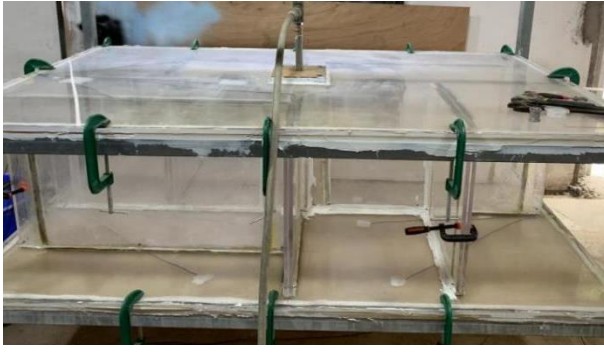

**Figure 5.** Model Assembly and Forming Diagram.

### 3.3. Analysis of Transport Mechanism

The rock mass is a discontinuous body divided by fractures, and the slurry will migrate through the network of fractures in the rock mass. The slurry in the field test device exhibits the migration law after it has been injected into the drill pipe: firstly, they penetrate the horizontal fracture at the bottom and propagate around it. After filling and saturation of the lower horizontal fracture, the slurry penetrates the vertical fracture and continues to

migrate upward. When a vertical fracture fills, it penetrates the upper horizontal fracture and spreads around the fracture. The slurry then infiltrates, migrates, and diffuses as it propagates to the vertically connected fractures. The transport of the grouting slurry through the jointed bedded rock mass shows a cyclic and gradual process of infiltration, diffusion and backfilling, vertical ascent, horizontal diffusion, and infiltration, capping, and backfilling. The experimental on-site transport process is shown in Figure 6.

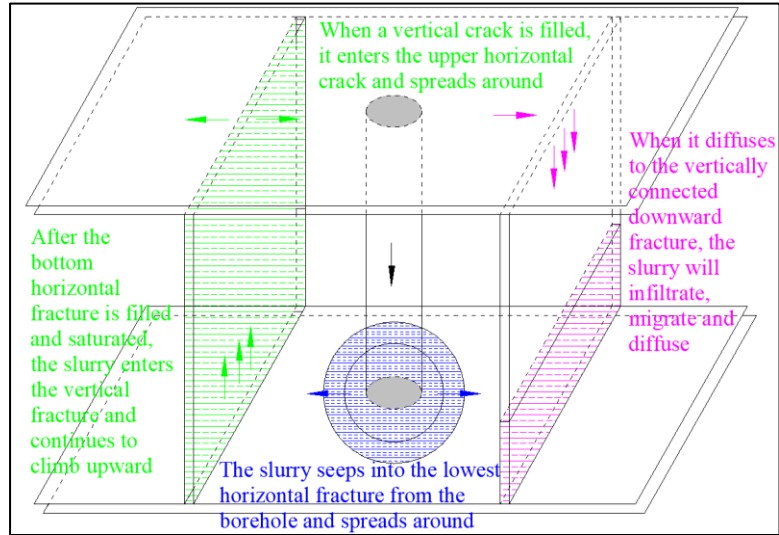

**Figure 6.** Schematic diagram of grouting slurry transport mechanism in a layered jointed rock mass.

### 3.4. Analysis of Slurry Diffusion Morphology

The initial diffusion of the slurry in the bottom horizontal fissure can be regarded as each homogeneous fluid, and a large number of scholars have researched that the diffusion of slurry in the horizontal fissure is radial ring diffusion centered on the grouting hole, and the trace of the slurry front is circular. The height of the slurry diffusion in the vertical fissure decreases with the increase in the horizontal diffusion radius at the beginning of the slurry diffusion in the vertical fissure by the influence of self-weight and fissure resistance, and its form is approximately a parabola with the grout hole as the apex and the opening downward, as shown in Figure 7. This experimental phenomenon is consistent with the conclusion derived from the slurry flow control equation.

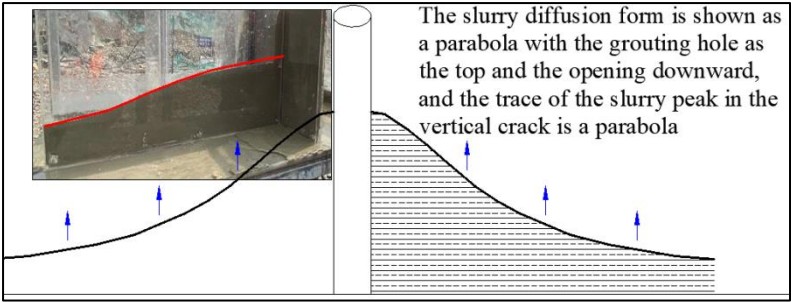

**Figure 7.** Initial morphology of slurry diffusion in vertical fractures.

### 3.5. Analysis of the Mechanism of Slurry Deposition and Nucleation Sealing in Fractures

There are two stages in the grouting process: "filling" and "saturation". During the filling phase, the slurry penetrates and fills the majority of the fractures. During the saturation stage, the excess water in the slurry produces a Terzaghi-like mechanical consolidation phenomenon of the soil at the highest saturation pressure and is discharged, causing the cement particles to pile up in close proximity to each other. The flow velocity and pressure of the slurry entering the rock fracture rapidly decrease as the distance from

the borehole increases. When the flow rate of slurry through the fracture decreases to a certain critical value, once the critical flow rate is reached, the cement particles initially sink to the bottom under gravity. Sedimentation leads to a reduction in the cross-sectional area of the grout, which in turn leads to changes in the pressure and flow rate of the grout. The excess water flows in the form of clear water into the small gaps at the top of the sedimentary layer until the fractures are completely filled.

Smaller particles of cement form larger polymers as a result of flocculation and hydration reactions, thereby blocking the seepage channel and affecting critical fracture opening. The deposition of cement slurry occurs simultaneously with the diffusion of the slurry. Excess free water in fractures can be separated following cement particle deposition, but it is difficult to affect the deposited cement particles again unless there is a disturbance or the rate of water flow increases. The reason for this is that the surface area that can completely fill the entire fracture and achieves effective sealing and settling is much smaller than the diffusion range of the slurry. The region capable of depositing and filling the fracture and supporting the dynamic water impact of the fracture is called the sediment retention core, abbreviated as the "retention core" as illustrated in Figure 8.

It is crucial to note that the grouting and water-blocking effect of cement slurry is dependent on the diffusion range of the slurry retention core. Not only does the cement slurry in the retention core zone after water precipitation achieve complete crack filling, but it also forms a zone of obstructions that impede the movement of the water flow through the fractures, forcing the fracture flow field to change, forming flow around or restricted to the region centered on the core. During the sedimentation and diffusion process of the slurry core as the grout progresses, as shown in Figure 9, the effective deposition range of the retention core gradually increases, resulting in a gradual reduction in the width of the water channel.

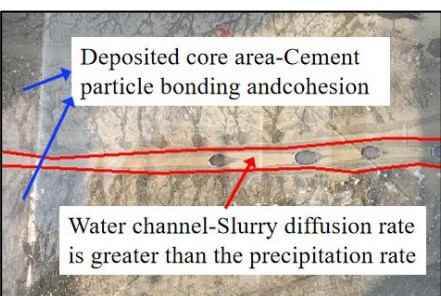

**Figure 8.** Slurry deposition and core sealing in horizontal fractures.

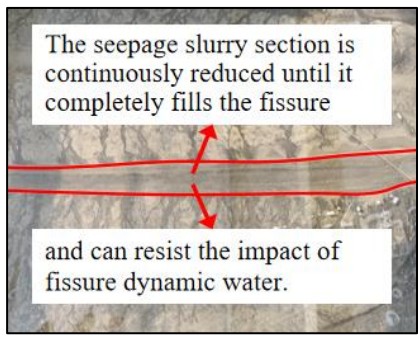

**Figure 9.** Sealing diagram of water channel.

Once the grouting has stopped, due to incomplete sealing of the horizontal fractures at the base, the slurry will gradually soften and sink into the vertical cracks due to the influence of its own weight, but will eventually retain a certain height, as shown in Figure 10, which is the effective sealing height for the slurry to settle into vertical cracks during the grouting process. As grouting proceeds, the height of sediment cores in vertical fractures will continue to increase, ultimately leading to the sealing of vertical fractures.

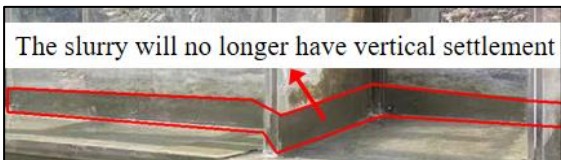

**Figure 10.** Slurry deposition and core sealing in vertical fractures.

## 4. Analysis of Water Blocking Effect in Field Industrial Tests

In order to verify the feasibility of implementing curtain grouting and the effectiveness of the curtain parameter design, and to provide reliable parameters for subsequent large-scale grouting, industrial tests of curtain grouting were conducted at representative locations on-site according to the relevant specifications. The design of the experimental section is based on the adoption of a pure water slurry for grouting, with a water–cement ratio of 0.6:1. The length of the experimental section is 50 m, with a borehole depth of 60 m. The double-row holes are arranged in a plum blossom shape, with a spacing of 5 m and 9 m. The grouting section is 20 m long, with a single injection rate of 600 kg/m. Taking a value of 1.7 Mpa, the pressure at the end of the grout is 2.5 times the net pressure of the water.

When the curtain of the experimental section forms, it will alter the groundwater flow field within a certain range in the vicinity, forming a relatively independent hydrogeologic unit. Geophysical methods and inspection hole detection are currently primarily used to evaluate and analyze the effectiveness of on-site grouting and water blockage. Zolfaghari et al. [30] developed a Q-logging scheme. The rock core drilled at the grouting site was used to calculate the modulus of deformation of the rock using experimental equations to study the impact of cement grouting on the rock properties; Fan et al. [31] proposed a comprehensive evaluation of the effectiveness of curtain grouting based on the results of WPTs, drilling and coring, and borehole television imaging.

### 4.1. Intuitive Water-Blocking Effect of Grouting

After grouting the experimental section, inspection holes are drilled into the interior of the curtain to check for a water-blocking effect. Given the sampling pattern of the inspection hole core in Figure 11 and the borehole imaging image in both the pre-and post-grouting boreholes in Figure 12, in addition to having some bond strength, it is evident that the slurry has a good filling and compaction effect on the open fractures in the interlayer and on the vertically connected fractures.

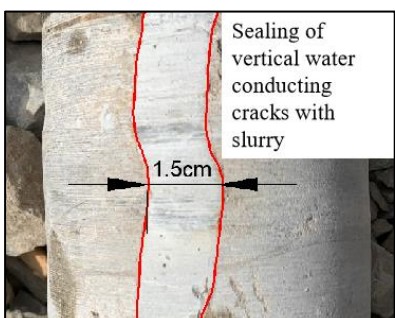

**Figure 11.** Inspection hole coring diagram.

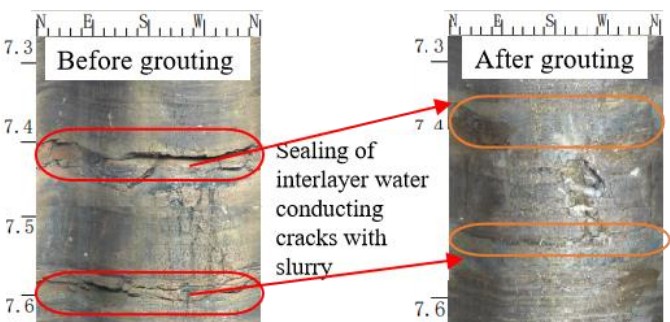

**Figure 12.** Imaging of boreholes in grouting holes before and after grouting.

*4.2. Analysis of Water Permeability*

The permeability of the formation is derived from the borehole piezometric test, which uses high pressure to press water into the borehole and calculates the in situ test to understand the fracture development and permeability of the rock based on the water absorption of the rock, calculated as follows:

$$q = Q/(p_t L) \tag{5}$$

where $q$ is the permeability, Lu; $Q$ is the injection volume, L/min; $p_t$ is the injection pressure, MPa; and $L$ is the length of the water injection test section, m.

A water pressure test is carried out on the grout section before each grouting in order to calculate the permeability of the rock layer. The frequency plot and cumulative frequency plot of the permeability rate are shown below.

From Figure 13, it can be seen from the frequency plot and the cumulative frequency plot of the permeability rate that the first-order hole grouting section is predominantly composed of high-permeability forming properties, whereas the second-order hole permeability rate is relatively weakened while the third-order hole exhibits low-permeability forming properties. The cumulative frequency plot of the permeability rate for each pore sequence is from right to left, followed by sequences I, II, and III. These results indicate that pores or fractures in the rock layer are being filled to some extent by the first-order hole grouting treatment, and the permeability of the rock layer is significantly enhanced by second-and third-order hole densification. As the grouting sequence progresses, the permeability rate shows a trend of gradually decreasing the number of high-value intervals and increasing the number of low-value intervals. The permeability of the second-order hole decreases by 40.9% compared to the first-order hole, and compared to the second-order hole, the third-order hole decreases by 69.4%. The third-order hole decreases by 81.9% relative to the first-order hole, showing a positive and significant decrease. As the grouting sequence progresses, the fractures are progressively filled in, and the effect of the grouting is advantageous.

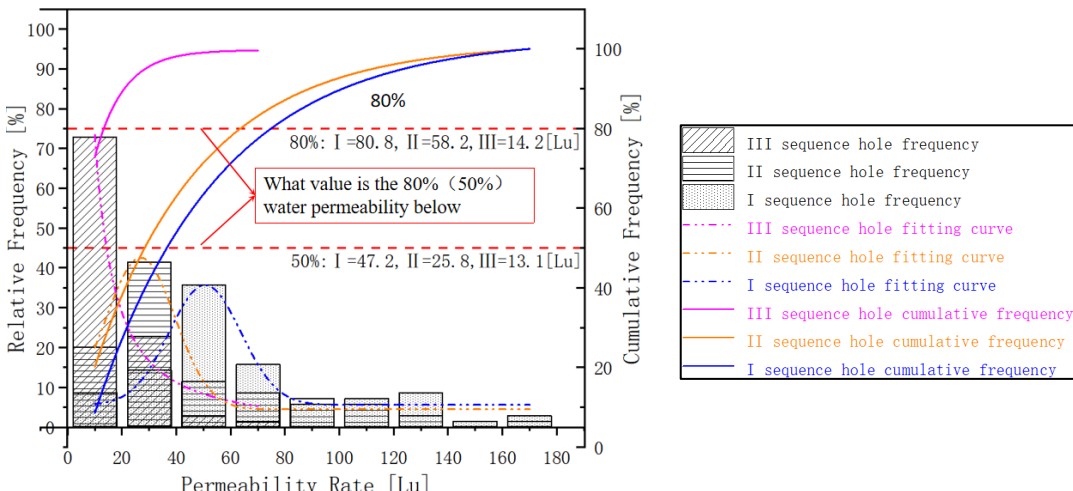

**Figure 13.** Permeability frequency curve and frequency accumulation curve.

### 4.3. Acoustic Testing Analysis

The use of ultrasonic testing technology to evaluate the effect of rock grouting reinforcement and water plugging is an effective and practical means. It measures the law of change in its internal acoustic wave velocity in different states of the original rock body before and after grouting, according to which the physical properties related to the original rock body are inferred. Since the longitudinal wave propagates faster than the transverse wave, and it is simpler and more applicable to test the longitudinal wave velocity in rock mass, it has the advantages of high testing accuracy, low testing cost and no damage to the specimen, so this test is conducted on the longitudinal wave velocity of rock formation [32].

Using the pair measurement method with butter as the propagation medium for various acoustic testing, as shown in Figure 14, the average wave speed of various rock block test results are: intact rock block 4.2 km/s, fractured rock block 3.8 km/s, fracture-filled rock block 3.5 km/s, and swept hole cement block 2.7 km/s.

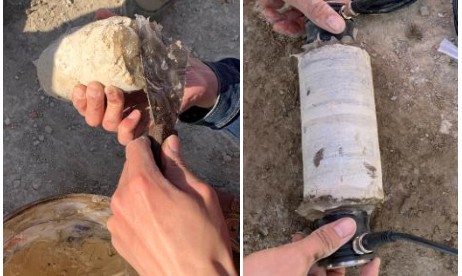

**Figure 14.** Field block wave velocity test diagram.

The sound waves in the hole before and after grouting were tested, and the patterns of wave velocities at different depths are as follows: within a depth interval of 10 m from the surface to the subsurface, the rock mass is relatively fragmentary, with some areas affected by the disturbance caused by the excavation. The wave velocity of the rock mass is relatively low, gradually increasing downslope, but not in a nonlinear fashion. A corresponding decrease in wave velocity is observed in the highly weathered and fractured section. After grouting, the wave velocity of the rock mass is significantly higher than before grouting, in particular, in the fractured region with developed fractures. After grouting, the wave velocity of the rock mass is significantly increased.

The acoustic medium in the grouted hole is water, so the tested wave velocity is somewhat different from the results of rock block pair measurements. From Figures 15 and 16, along with scatter plots of the wave velocity distribution before and after grouting, as can

be seen, the dispersion of the wave velocity distribution prior to grouting is relatively large, and the proportion of small ranges of wave velocities is also high. The wave velocity range with the widest data distribution is 2.54–2.94 km/s. There is a significant increase in wave velocity after grouting, and the distribution is concentrated in the relatively high wave speed range. The data decreases significantly or even disappear in the limit of the small wave velocity, with the largest distribution of data in the wave velocity range between 3.02 and 3.37 km/s. The peak wave velocity is 4.02 km/s, which is closer to the wave velocity of the intact rock mass. This result indicates that fracture filling after grouting improves the integrity of the stone mass, and grouting has a strong effect on the improvement of the overall performance of the rock mass.

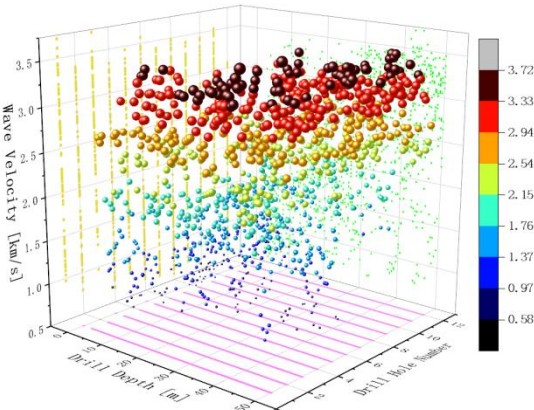

**Figure 15.** Scatter plot of wave velocity before grouting.

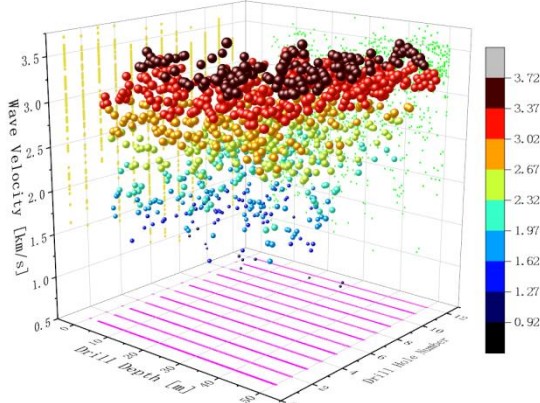

**Figure 16.** Scatter plot of wave velocity after grouting.

## 4.4. Analysis of Geophysical Effect Detection

After grouting is completed in the experimental section, both the high-density electrical method and the inter-hole electromagnetic wave CT method are employed to detect the water-blocking effect of the grout. Figure 17 shows the results of the high-density electrical tests with the measurement line arranged vertically along the curtain. Note that the resistivity of the curtain grout region is significantly higher than that of the region outside of the grout domain, with a width of about 30 m. Figure 18 shows the results of testing the inter-hole electromagnetic wave CT method. The two boreholes detected are positioned at the center and outside of the curtain, respectively, at a distance of 15 m and within a 13 m range of the centerline of the curtain. The electromagnetic wave exhibits clear weak absorption and is uniformly distributed. Combining the two detection results, fractures within a certain range can be seen after the grout has been filled with a cement slurry, resulting in a reduced hydraulic connection on either side of the curtain. The curtain may

play a good role in water splitting, and the effective sealing range of the diffusion of the slurry is in the range of 20 to 30 m.

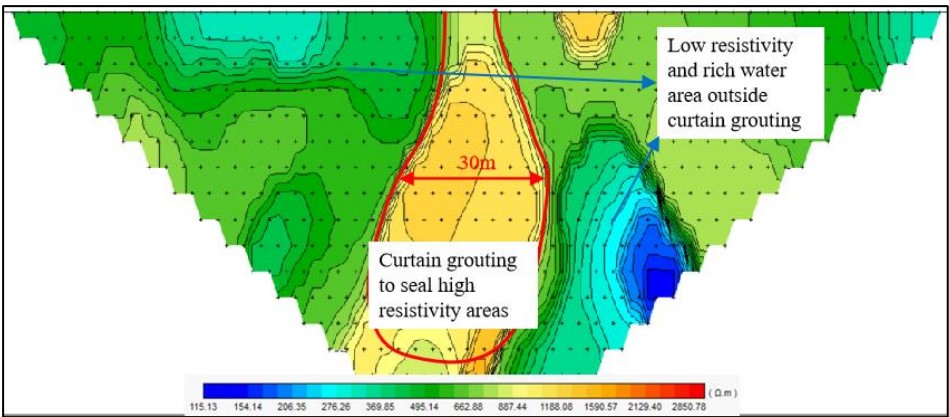

**Figure 17.** Inspection diagram of high-density electrical method curtain grouting effect.

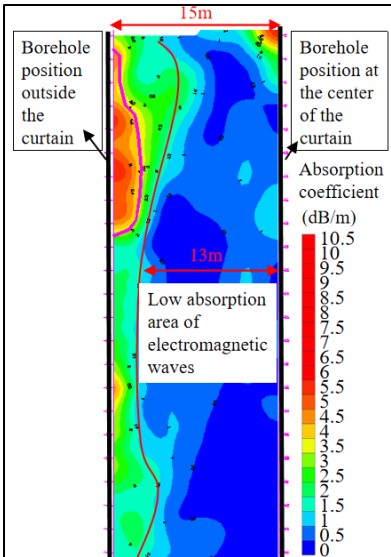

**Figure 18.** Inspection diagram of curtain grouting effect using electromagnetic wave CT method between holes.

## 5. Optimization of the Grouting System

The purpose of the experiment is to provide engineering support for the design and construction of large-scale curtain grouting in the future, as well as to obtain the necessary grouting data in advance. Niu et al. [33] used the permeability of the rock layers to predict the amount of unit grouting of the karstic curtain grout; Fu et al. [34] conducted field grouting tests and explored the spacing between the grout holes based on a reliability analysis; Liu et al. [35] explored an effective method for processing water-rich sand layers through curtain grouting in key areas, forming a comprehensive grouting treatment method.

This paper concludes that the water–cement ratio, grouting pressure, section length, unit grouting volume and drilling spacing of the grouting parameters are changed within a certain interval with the grouting process and the change in formation information, and are not a fixed value in the design. The whole grouting process is a dynamic control process. The reasons for parameter adjustment are as follows:

The water–cement ratio is divided into open tank water–cement ratio and main filling water–cement ratio. To ensure the slurry can be fully diffused at the initial stage, the diluted

slurry must be used to flush the fissures, and at the later stage, the fissures must be sealed with a high-concentration slurry; a suitable water–cement ratio can be selected according to the water permeability of the rock formation.

The pressure of the slurry injection initially fluctuates greatly; due to the development of fissures that may not be under pressure, the slurry can spread by its own weight, or the pressure is large at the beginning and suddenly smaller in the middle because the fissures are swept open, therefore, when the permeability of the rock formation is large, there is a need to reduce the pressure to avoid wasting the slurry, and when the permeability is small, there is a need to increase the pressure to make the slurry fully spread.

If the length of the grouting section is too long, the pressure of the grouting pressure at a certain point of the borehole is bound to decrease, which will affect the diffusion radius of the slurry and affect the quality of the grouting by failing to reach the designed thickness of the curtain wall. If the drilling hole has serious water leakage and the water does not return to the hole in the drilling process, the section length needs to be reduced. If the core is intact, the section length can be increased appropriately.

Grouting tests with different drilling spacing were added on site, and it was found that when the drilling spacing was greater than six meters, the permeability of the curtain body was greater than 10 Lu using triangular and rectangular hole layouts, which could not meet the water plugging requirements.

After dynamic optimization and adjustment of the above parameters, an optimal unit grouting volume was obtained for different stratigraphic permeability, and the parameters were diversified according to the characteristics of the injected strata, as shown in Table 1.

**Table 1.** Grouting parameters and process optimization table.

| Parameter | Design Value | Optimization Value | | |
|---|---|---|---|---|
| | | **Permeability Rate of Rock Layer before Grouting (Lu)** | **Water–Cement Ratio for Pre-Filling** | **Water–Cement Ratio for Main Filling** |
| Slurry water–cement ratio | 06:1 | 10~20 | 5:1 | 1:1~2:1 |
| | | 21~50 | 3:1 | 0.8:1~1:1 |
| | | 51~100 | 2:1 | 0.8:1 |
| | | >100 | 1:1 | 0.6:1 |
| Grouting section length | 20 m | Permeability rate of rock layer before grouting (Lu) | Grouting section length (m) | |
| | | 10~20 | 10 | |
| | | 21~50 | 8 | |
| | | 51~100 | 5~8 | |
| | | >100 | 5 | |
| Grouting pressure | 2.5 times the hydrostatic pressure, with a final pressure of 1.7 MPa | Permeability rate of rock layer before grouting (Lu) | Final grouting pressure (MPa) | |
| | | 10~20 | 2.4 | |
| | | 21~50 | 1.7~2.4 | |
| | | 51~100 | 1.7 | |
| | | >100 | <1 | |

**Table 1.** *Cont.*

| Parameter | Design Value | Optimization Value | | |
|---|---|---|---|---|
| | | **Permeability Rate of Rock Layer before Grouting (Lu)** | **Water–Cement Ratio for Pre-Filling** | **Water–Cement Ratio for Main Filling** |
| Unit grouting volume | 600 kg/m | Permeability rate of rock layer before grouting (Lu) | Optimal single injection rate (kg/m) | |
| | | 10~20 | 330 | |
| | | 21~50 | 440 | |
| | | 51~100 | 600 | |
| | | >100 | 1100 | |
| Row spacing and hole layout between drilling holes | Double row holes, arranged in a plum blossom shape, with a spacing of 5 m and a row spacing of 9 m | Triangular layout method, hole distance 6 m, rock fracture development area shortened to 5 m, or in the middle of the triangle to play encryption holes. | | |

## 6. Conclusions

By combining similar simulation experiments with on-site industrial experiments, it was possible to analyze the feasibility and water-blocking effect of using curtain grouting for the prevention and control of water in large water deposits. The results of this study provide technical support for the sustainable mining of ore bodies below the groundwater level in the open pit at Jianshan Phosphate Mine.

1. The transport of the grouting slurry through the jointed bedded rock mass shows a cyclic process of infiltration, diffusion and backfilling, vertical ascent, horizontal diffusion, and infiltration, capping, and backfilling. Once the slurry deposits and retains the core, it can play a sealing role.
2. The height of slurry diffusion in vertical fissures decreases with the increase in the horizontal diffusion radius, and the form of the diffusion front is approximated as a parabola with the grouting hole as the axis and opening downward.
3. Physical means can detect the filling of the fissure, and the effective sealing range of slurry diffusion in the field test is about 20–30 m.
4. Field grouting parameters are not fixed values but need to be adjusted dynamically according to the actual site to obtain a complete set of diversified value-taking grouting systems.

**Author Contributions:** Investigation: Y.Z. and K.H.; curtain grouting technology: Y.Z., K.H. and W.L.; test model: Y.Z. and W.L.; grouting simulation test: Y.Z., W.L. and H.S.; industrial field test: Y.Z., K.H., W.L. and H.S.; water permeability test: Y.Z. and H.S.; acoustic test: Y.Z. and W.L.; geophysical detection: Y.Z., W.L. and H.S.; writing-review and editing: Y.Z., K.H., W.L. and H.S.; conceptualization: K.H.; resources: K.H.; supervision: K.H. All authors have read and agreed to the published version of the manuscript.

**Funding:** The work described in this paper received financial support from the Guizhou Provincial Science and Technology Projects (No. QKHZC [2021] General 407).

**Institutional Review Board Statement:** Not applicable.

**Informed Consent Statement:** Not applicable.

**Data Availability Statement:** Not applicable.

**Conflicts of Interest:** The authors declare no conflict of interest.

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
