# Peer review of "Research on Sustainable Mining and Water Prevention in Large Open-Pit Water Deposits"

_sustainability, doi:10.3390/su151310238_

Round 1

Reviewer 1 Report

Based on the research background of water control in Jianshan phosphate mine, the author analyzed the diffusion and plugging mechanism of grouting, optimized the grouting system and parameters, and provided technical support for sustainable mining of open-pit mines affected by flooding. The research has certain application value.This work could be accepted after minor revisions. Other questions were shown below:

1. The form style in the paper is incorrectly referenced, so it needs to be redrawn and adjusted to beautify the content;

2. As shown in Figure 17, the results of electromagnetic wave CT detection between holes lack cloud map color codes, and the values are not clear, which need to be supplemented;

3.The text description in some drawings is too complicated, please simplify and highlight the test resultsï¼›

4. In the fifth part of the text, the final optimization results are directly presented, and the value basis of parameters should be described to make the logic of the article more rigorous;

5.The conclusion part should be more refined to make the findings and contributions of the paper clearer. Furthermore, please note the difference between the conclusion and abstract.

Reviewer 2 Report

This paper focuses on sustainable mining and water prevention in large open-pit phosphate deposits. The self-developed three-dimensional visualization model of jointed rock mass grouting was used to conduct research on the mechanisms of slurry transport, diffusive morphology, and sealing. The findings revealed that after adopting the curtain grouting technology, the water-conducting cracks in the formation can be properly filled and sealed and have a proper water-blocking function. However, further revisions prompt a better understanding of the manuscript. The reviewer would like to suggest a minor revision. The main concerns are:

(1) L.47–69: The authors should explain more clearly the novelty, research gap and contribution of this present work. Indeed, the last paragraph of the introduction part should be revised and rewrite according to this comment.

(2) In Section 3.1, the description of the test setup (method) is not adequate and the test system is not elaborated; more details should be supplemented, such as the design and process of grouting. In addition, it is suggested to add some relative pictures of the test details.

(3) The author conducted a water permeability analysis in Section 4.2, it is suggested to give more details (e.g., permeability testing methods, process, etc.) of the testing.

(4) In Section 4.3, the acoustic test analysis is carried out, and it is necessary to describe the method, process, and experimental equipment of the acoustic test in detail.

(5) The data labeled in Figure 2 should specify the unit. Figure 10 should be clearer and more readable.

(6) The value of resistance absorption rate in Figure 17 is not clear, so add the value represented by each color.

(7) The number of references can be increased as appropriate.

(8) Please ensure that the form format meets journal publication requirements.

(9) The conclusion requires highly refined language, precise wording, clear and specific text.

(10) Please ensure that the reference format meets the journal publication requirements.

Minor editing of English language required

Reviewer 3 Report

sustainability-2407392-review

Research on Sustainable Mining and Water Prevention in Large Open-Pit Water Deposits

In this work, the author independently developed a set of 3D visual crack grouting model, which has certain novelty. In addition, industrial tests were carried out in the field, and the effect of curtain grouting on water plugging was verified by various means, and a set of relatively complete grouting parameters were obtained, which can provide guidance for sustainable mining of open pit large water deposit. This work could be accepted after minor revisions. Other questions were shown below:

1. Water inrush occurs mostly in coal mines, which is one of the most threatening disasters in coal mine production process. Many scholars have carried out a great deal of water prevention and control work in underground coal mines, suggesting that it be referenced for reference.

2. The test device is independently developed, so it should be introduced in more detail to highlight its innovation.

3. Adjust the text in Figures 8 through 10 to make it concise and attractive.

4. In Figure 13, 50% and 80% of the line meanings should be written.

5. In the conclusions, in addition to summarizing the actions taken and results, please strengthen the explanation of their significance.

Minor editing of English language required

Reviewer 4 Report

1-      I could not find the novelty of the manuscript. The authors should review more theoretical aspects of the subject and clarify their contribution to this research

2-      What is the logic behind the design diagram of the three-dimensional visualization layered jointed rock mass grouting model? How have the authors defined the space between the cracks?

3-      Fig. 5 is not clear? It is clear that the diffusion pattern of the slurry

4-      The parabolic shape and 8 morphology of slurry diffusion are not precise and they need quantitative measurements.

5-      Reproducibility of the results for the permeability frequency curve and frequency accumulation curve is required

6-      These sentences are qualitative and some quantitative measurements are required to approve them “the dispersion of the wave velocity distribution prior to grouting is relatively large” or “grouting has a strong effect on the improvement of the overall performance of the rock mass”

7-      Fig. 17 requires the legend to show the different color values.

8-      It is unclear how the authors concluded the Optimized grouting system as shown in Table 1.

I could not find any special issue on English Language writing
